# Optimization of S-Nitrosocaptopril Monohydrate Storage Conditions Based on Response Surface Method

**DOI:** 10.3390/molecules26247533

**Published:** 2021-12-13

**Authors:** Lingyi Huang, Yu Zhou, Yizhi Wang, Min Lin

**Affiliations:** 1School of Pharmacy, Fujian Medical University, Fuzhou 350122, China; lingyi.huang@fjmu.edu.cn; 2Key Laboratory for Chemical Biology of Fujian Province, MOE Key Laboratory of Spectrochemical Analysis and Instrumentation, College of Chemistry and Chemical Engineering, Xiamen University, Xiamen 361005, China; zhouyu_921@163.com; 3Nanjing Leechdom Biopharm Technology Co., Ltd., Nanjing 211500, China; 4School of Intelligent Science and Control Engineering, Jinling Institute of Technology, Nanjing 211169, China; w_yz@jit.edu.cn; 5College of Materials and Chemical Engineering, Minjiang University, Fuzhou 350108, China

**Keywords:** S-nitrosocaptopril monohydrate, temperature, nitrogen purity, deoxidizer

## Abstract

From unstable crystals to relatively stable monohydrate crystals, many researchers have been working on S-nitrosocaptopril for more than two decades. S-nitrosocaptopril monohydrate (Cap-NO·H_2_O) is a novel crystal form of S-nitrosocaptopril (Cap-NO), and is not only a nitric oxide (NO) donor, but also an angiotensin-converting enzyme inhibitor (ACEI). Yet, a method for long-term storage has never been reported. In order to determine the optimal storage conditions, Plackett–Burman (PB) design was performed to confirm the critical factors. Response surface methodology (RSM) was employed to determine the optimal Cap-NO·H_2_O storage condition, based on the rough interval determined by the path of steepest ascent experiment. The optimized storage condition was denoted as nitrogen purity of 97%, temperature of −10 °C and 1.20 g deoxidizer. In this case, a final preservation rate of 97.91 ± 0.59% could be obtained. In specific storage conditions, Cap-NO·H_2_O was found to be stable for at least 6 months in individual PE package, procreating a potentially applicable avenue.

## 1. Introduction

S-nitrosocaptopril is a S-nitrosylated captopril developed more than 20 years ago [1], but its technical bottleneck of large-scale synthesis was recently overcome [2]. Moreover, S-nitrosocaptopril monohydrate (Cap-NO·H_2_O) is a novel and stable compound based on S-nitrosocaptopril (Cap-NO), which is not only a nitric oxide (NO) donor but also an angiotensin-converting enzyme inhibitor (ACEI). Cap-NO·H_2_O has great potential for the study and possible treatment of numerous diseases such as pulmonary hyperattention [2,3,4], hypertension regardless of renin status, angina pectoris [5] and congestive heart failure [6], acute respiratory distress syndrome [7] and preventing postoperative circulation tumor cells metastasis [8].

One of the most common ways of degradation of S-nitrosocaptopril Monohydrate (Cap-NO·H_2_O) during storage is oxidation reaction [9]. When Cap-NO·H_2_O loses crystal water, the functional groups of Cap-NO would exist in three forms: [-S-N = O ←→ -S+ =N-O- ←→ -S-/N≡O+]. Although the content of -S-/N≡O+ is only 6–10%, it will attack the -S-N bond of Cap-NO, resulting in a knock-on effect of Cap-NO degradation [2]. At the same time, if there are oxygen molecules in the environment, oxygen free radicals will be generated and interact with Active Pharmaceutical Ingredients (API), impurities and other auxiliary materials to generate more free radicals and accelerate the degradation of Cap-NO·H_2_O. Although the precise mechanism of the interaction of the oxidation process is not very clear, the reaction process is generally an auto-oxidation process [10], including three processes of initiation, circulation and termination.
Initiation:In· + R-H→R· + In-HCirculation:R· + O_2_→ROO·
R· + R-H→R-H·
R-OO·+ R-H→ROOH + R·Termination:2ROO→Product

Among them is an unknown free radical initiator, and R-H represents pharmaceutical substances, excipients, or other pollutants. Whether directly reacting with the drug or indirectly acting through other excipients, oxygen molecules participated in the propagation step and the catalytic cycle of the oxidative degradation of the pharmaceutical substance in the drug formulation. The oxidative degradation of API leads to a decrease in drug efficacy and an increase in oxidative degradation products.

The oxidative degradation of API leads to the decrease of efficacy and the increase of oxidative degradation products (impurities). The rate of oxidative degradation depends on the nature of API and the sensitivity of storage conditions (such as temperature, humidity, oxygen concentration, light and time, etc.). It is known that the harmful effects of the oxidation process of Cap-NO·H_2_O include product discoloration (from red to white), changes in dissolution rate, red-brown gas generation and bad odors, etc. [2]. The key point is that the generated oxidative degradation products may lead to the adverse reactions of API, or the effects on the efficacy of the medicine.

To solve this problem, the common solutions include: (1) adding non-mixed antioxidants or chelating agents and (2) placing it in an inert gas atmosphere [11]. On this basis, the API is stored in a constant and low temperature environment. Although the form of adding antioxidants or chelating agents to drugs is more common, there are fewer studies on the control and application of multi-influencing factors under conditions of nitrogen content. This may be because it is difficult to quantitatively evaluate the oxygen content of a packaging system. However, inert gas protection is a direct method to effectively isolate the API and its auxiliary materials from oxygen. By removing one of the two key reactants in the oxidation cycle from the packaging, the degradation of susceptible formulations can be minimized, and the shelf life of oxygen-sensitive commodities will be prolonged.

PBD is an experimental program developed by R.L. Plackett and J.P. Burman in 1946 [12]. In order to improve the quality control process, the design could be used to study the influence of design parameters on the system state, so as to make reasonable decisions. The orthogonal array designed by Plackett and Burman (PB) can produce unbiased estimates of all major influences in the least possible design. However, the initial estimate of the optimum values derived from the fractional factorial design could be far from the true optimum values. Thus, the method of steepest ascent was usually introduced to move rapidly to the general vicinity of the actual optimum [13]. Then, response surface methodology (RSM) should be conducted to obtain an optimal response as a statistical multi-response optimization method [14]. RSM is an effective statistic tool with minimized experimental trials and a 3D description of the interactions of parameters [15].

In the present paper, the Plackett–Burman method was performed to screen significant impacts on API storage out of five factors based on previous studies on Cap-NO·H_2_O stability. Then, the response surface method was used to obtain the optimal storage conditions of the small test level that provided guidance for the storage conditions of the industrial scale-up stage.

## 2. Results

### 2.1. Plackett–Burman Experimental Results and Analysis

As indicated in Figure 1, Cap-NO·H_2_O (10 μg/mL) was observed as eluted at the wavelength of 215 nm, whose retention time was 9.65 min.

The experimental results based on the Plackett–Burman experimental grouping design were shown in Table 1.

The results of ANOVA analysis are shown in Appendix A. The significance of the model (*p* value) was below 0.0001, which indicated that the experimental model was significant; namely, the model had a high degree of fit and was suitable for predicting the importance of each factor. The order of the influence of each influencing factor on the preservation rate of Cap-NO·H_2_O was “*X*_4_ > *X*_3_ > *X*_2_ > *X*_5_ > *X*_1_”; namely, the influence of each factor on the preservation rate of Cap-NO·H_2_O was as follows: Temperature > Nitrogen purity > Deoxidizer > Desiccant > Light. The significance of the temperature was below 0.0001, which indicated that the temperature had a very significant effect on the preservation rate of Cap-NO·H_2_O. The significance of the nitrogen purity of the storage atmosphere was below 0.05, which indicated that the nitrogen purity had a significant impact on the preservation rate of Cap-NO·H_2_O. While the significance of deoxidizer was 0.0596 (larger than 0.05), which was not statistically significant, its value was near the critical value. It may be that the number of repetitions was not enough, so it should be considered as one of the key factors.

According to the residual shown in Figure 2a, the residual value was calculated based on the difference between the predicted value and the actual value. In this figure, the actual residual value was plotted along the horizontal x-axis, while the predicted expected value was plotted along the y-axis. It could be seen that basically all the values fell near the straight line, indicating that the residuals obeyed a normal distribution that could intuitively support the results of the above model analysis. As shown in Figure 2b, the Pareto chart ranked the absolute values of each factor’s standardized effects in descending order. Among them, only the temperature was above the significant line, the nitrogen purity was above the significant line, and the other factors and virtual factors were not significant, which could directly support the model analysis results.

### 2.2. Results and Analysis of the Path of Steepest Ascent

According to the results of the PB experiment, the insignificant factors (desiccant and light) were excluded. A climbing experiment was performed on the remaining factors: temperature, nitrogen purity, and deoxidizer. The experimental results were shown in Table 2. According to the maximum condition of the experimental results, the storage temperature of the center point of the response surface experiment was −8 °C, the nitrogen purity was 95%, and the deoxidizer was 1.13 g.

### 2.3. Results and Analysis of Response Surface Experiments

According to the response surface experimental design grouping, the results of the preservation rate of Cap-NO·H_2_O were shown in Table 3.

Based on the above experimental results, ANOVA was used for statistical analysis, and the results were shown in Appendix A. According to the statistical results, a ternary quadratic regression equation was fitted and shown as follows:(1)$$Y=-665.58008-0.12040X_{1}+15.06603X_{2}+62.06200X_{3}\;-4.08333X_{1}X_{2}+0.031667X_{1}X_{3}-0.35467X_{2}X_{3}\;-0.023755X_{1}{}^{2}-0.076130X_{2}{}^{2}-11.70311X_{3}{}^{2}$$

The R^2^ of the model was 0.9843, the adjusted R^2^ was 0.9531 and the expected R^2^ was 0.8162. The difference between the expected R^2^ and adjusted R^2^ was less than 0.2, indicating that the model was reasonable. From the analysis results in the table, it could be seen that the significance of the model was 0.0001, which was extremely significant, and the lack-of-fit term was 0.148145, which was not significant, indicating that the model could better fit and predict the experimental results. Under the interaction of multiple factors, the P value of temperature, nitrogen purity, and the deoxidizer were all less than 0.01, indicating that those three factors have a very significant impact on the storage of Cap-NO·H_2_O. The P values of nitrogen purity and deoxidizer were less than 0.05 in the two interaction factors, indicating that the interaction between nitrogen purity and deoxidizer had a significant influence in this storage model. However, the interaction between temperature and the other two factors was not significant in this model. As shown in Figure 3, the contour plot and the 3D surface of the center point, which was set by the climbing experiment, deviated from the actual experimental results. However, the equation had a stable point, when *X*_1_ was 10.0417, *X*_2_ was 96.4796 and *X*_3_ was 1.17574, then the output *Y* was 96.4796. Namely, when the stable point conditions were storage temperature of −10.0417 °C, nitrogen purity of 96.4796% and deoxidizer of 1.17574 g, then the predicted preservation rate (*Y*) was 96.4796. Meanwhile, the maximum value of *Y* was 98.296, when *X*_1_ was 10.042, *X*_2_ was 96.480 and *X*_3_ was 1.176, then the output *Y* was 98.296. When storage temperature was −10.042 °C, nitrogen purity was 96.480% and deoxidizer was 1.176 g, the predicted preservation rate (*Y*) was 98.296. From the perspective of the steepness of the contour, the temperature had the greatest influence on the response of the system, followed by the nitrogen purity and the deoxidizer.

According to the prediction of the model, the actual experimental conditions were adjusted to storage temperature of −10 °C, nitrogen purity of 97%, and deoxidizer of 1.20 g. Under this condition, three sets of repetitions were performed, and the final true value was 97.91 ± 0.59%, as shown in Table 4, which was basically consistent with the predicted value. Those results show that the model had certain guiding significance for the practical preservation of Cap-NO·H_2_O.

## 3. Discussion

Through the PB experiment, it was known that the key influencing factors for the preservation of Cap-NO·H_2_O were temperature, nitrogen purity of the storage environment, and the deoxidizer. However, the desiccant and light were not the key factors, which seems to differ from the known properties of the functional group –SNO [16]. The possible reasons for this result were as follows: (1) the experimental result may come from the ultraviolet light, which is the key factor for photosensitivity of Cap-NO·H_2_O [17], and the PA/PE composite sealing packaging bag may block ultraviolet light; and (2) the Cap-NO·H_2_O used in the experiment had low moisture content after drying, so it was difficult to obtain moisture from the environment under the sealed conditions of small space and relatively stable ambient temperature. During the experiment, the use of desiccant had no significant influence on the stable storage of Cap-NO·H_2_O crystals.

Three-dimensional (3D) response surface and two-dimensional (2D) contour plots were provided as graphical representations of the regression equation (Figure 3). As shown in Figure 3a,b, the effect of temperature on storage changed more rapidly than that of deoxidant and nitrogen purity, since the curved surface of temperature was much steeper than the curved surface of the latter. The factor of nitrogen purity only had a relatively large effect at lower levels of purity with steep and sharply dissected slopes (Figure 3a,c), while the change of deoxidant in intervals tends to be gentler with gradual slopes (Figure 3b,c). The elliptical shape of all the contour plots showed that three pairwise interactions were significant, particularly when the factor of temperature was included (Figure 3b,d).

The results of the response surface showed that under the interaction of temperature, nitrogen purity and deoxidizer, all three factors had a significant effect on the stable storage of Cap-NO·H_2_O crystals, but only the interaction of nitrogen purity and deoxidizer were statistically significant (Appendix A). The reasons were as follows: (1) temperature was a key factor. When the ambient temperature was high, the impurity coming from the synthesis of Cap-NO·H_2_O or the -SNO group was relatively active, and the form content of -S- /N≡O+ would increase and attack the functional groups of other molecules, causing chain degradation reactions. At this time, the control of the oxygen content and humidity of the environment would no longer suppress the occurrence of this process. Moreover, (2) the interactive relationship between the nitrogen purity and the deoxidizer were remarkable because when the deoxidizer absorbed the oxygen in the environment, the nitrogen purity in the circumstance increased relatively.

However, when the use of deoxidizer increased, the preservation rate of Cap-NO·H_2_O crystals would decrease. This may be because the selected mixed deoxidizer was a mixture of iron (Fe) powder. Although there was packaging to isolate Cap-NO·H_2_O crystals and deoxidizer, when the amount of deoxidizer is too high, some of it may exist as aerosol ions in the packaging environment, and metal ions could degrade the functional group of Cap-NO·H_2_O-SNO.

In summary, in the subsequent pilot trials and expanded production of drugs, due to the full consideration of the packaging factors of Cap-NO·H_2_O and the temperature control in the storage and transportation links, the common materials PA/PE can block ultraviolet radiation. The composite material can play a better role in blocking UV degradation of Cap-NO·H_2_O. The drying process of Cap-NO·H_2_O was an important step in the production process. This step can greatly save the cost of subsequent drug storage. At the same time, we could further consider replacing the deoxidizer or improving the packaging of the deoxidizer, so that the package would not only isolate the impact of deoxidizer on the API, but also deoxidize the environment, or replace the non-metal ion type deoxidizer. On the other hand, the experimental results of the response surface also gave us an inspiration of the application level, namely, in this trial with the deoxidizer, the high purity of nitrogen in the nitrogen-filled atmosphere was not a necessary condition.

## 4. Material and Methods

### 4.1. Chemicals and Reagents

Captopril (CAP, ≥98%) was purchased from Changzhou Pharmaceutical Co., Ltd. (Changzhou, China); sodium nitrite (NaNO_2_, AR) was purchased from Xilong Chemical Co., Ltd. (Shantou, China); sodium chloride (NaCl, AR), sodium hydroxide (NaOH, AR), calcium chloride (CaCl_2_, AR) and Vaseline (Paraffin Waxes and Hydrocarbon Waxes, C_21_H_27_NO_3_, AR) were purchased from Sinopharm Chemical Reagent Co., Ltd. (Shanghai, China); disodiumedtadihydrate (EDTA-2Na·2H_2_O, AR) was purchased from Gen View Company (USA); hydrochloric acid (HCl, 36~38%) was purchased from Lanxi Xuri Chemical Co., Ltd. (Lanxi, China); methanol (MeOH, CH_3_OH, LC) and phosphoric acid (H_3_PO_4_, LC) were purchased from Merck (Darmstadt, Germany); deoxidizer (30-type) and desiccant (30-type) were purchased from Hangzhou Lvyuan Fine Chemical Co., Ltd. (Hangzhou, China).

### 4.2. Instrumentation

Ultra-pure water system (MINI8-Y, Kertone, Northamptonshire, UK); vacuum packaging bag (PA/PE composite material, specification 7 cm × 10 cm, Hebei Wangshi Packaging Co., Ltd., Shijiazhuang, China); constant temperature and humidity/high and low temperature test chamber (JHY-H-80L, Xiamen Jinheyuan Technology Co., Ltd., Xiamen, China); cantilever mixer (LC-OES-60, Shanghai Lichen Instrument Technology Co., Ltd., Shanghai, China); circulating water vacuum pump (SHZ-D (III), Gongyi Yuhua Instrument Co., Ltd., Gongyi, China); high performance liquid chromatography (2695–2489, Waters, Milford, MA, USA); small multifunctional continuous vacuum pumping/inflating packaging machine (LF1080B, Zhejiang Dingye Machinery Equipment Co., Ltd., Zhejiang, China); medical Double Gauge Pressure Gauge/Reducing Valve (RH-002) Jinan Ronghai Medical Equipment Co., Ltd. Statistical Software: Design Expert 10 and IBM SPSS Statistics 23.

### 4.3. The Synthesis of Cap-NO·H_2_O

The synthesis of Cap-NO·H_2_O refers to the experimental procedures of published paper [2].

### 4.4. The Preparation and Use of Nitrogen Environment

A specific ratio of nitrogen was prepared by the improved drainage method [18]. As shown in Figure 4a, we filled the graduated gas cylinder with deionized water and buckled it upside down into a vessel containing deionized water. According to the ratio requirement of the experimental group, the corresponding gas with a certain volume ratio was filled from the gas charging port, and the corresponding volume of water was discharged to obtain the nitrogen-containing gas with a corresponding volume ratio.

The calculation method of charging gas was shown in the following formula:(2)$$\frac{78\%x+y}{x+y}=Q\%$$
where:

*x*—Response volume of air; *x* ≥ 0.

*y*—Response volume of nitrogen; *y* > 0.

*Q*—The percentage of nitrogen in the gas mixture; *Q* > 78.

After a simple transform, the equation can be denoted as follows:(3)$$\frac{x}{y}=\frac{100-Q}{Q-78}$$

After the gas collection was complete, we placed the bottle body upright. We put glass wool on the bottom of the drying tower to prevent the desiccant above the waist from falling, filled the drying tower with CaCl_2_ desiccant above the waist, then applied petroleum jelly to the upper bottle and rotated it to prevent air leakage, and aligned the air outlet with the tower body. The connecting port was used to prevent the moisture from affecting the API in the package. Deionized water was injected from the water injection port, and the nitrogen-containing gas was discharged and injected into the API package. The specific link method is shown in Figure 4b.

### 4.5. Nitrogen-Filled Packaging Method

Nitrogen filling and sealing adopted a small multifunctional continuous vacuum pumping/inflating packaging machine (LF1080B). The operating parameters were as follows: power was 1.15 kW, power source was 220 V 50 Hz and sealing speed was 0–12 m/min. A total of 2 g Cap-NO·H_2_O was weighed and placed into the packaging bag. The procedure was as follows:(1)Nitrogen filling: connected the pressure control valve to the nitrogen cylinder (tighten), kept the pressure regulating valve closed, turned on the cylinder switch, and adjusted the pressure to 0.5–1 kPa. Turned on the packaging machine and the power switch, then turned on the heating switch to set the heating temperature to 220 °C. When the temperature reached the set temperature, turned on the fan, air extraction and nitrogen flushing. Adjusted the nitrogen inlet pressure valve to make the pressure appropriate (about 0.1–0.3 kPa).(2)Mixed gas filling: connected the drying tower and the ultrapure water machine system to the water injection port for continuous water injection.(3)Charging air: did not connect the air port.

Packing: Inserted the suction tube into the side of the packaging bag containing the sample (away from the conveyor belt), then stepped on the foot switch to start pumping. After the suction was completed, moved the suction tube to the conveyor belt and kept for 1–3 s until it was full of gas, so that it could enter the conveyor belt for sealing.

After packaging, we adjusted the set temperature to room temperature (without turning off the fan), and turned off the air extraction and inflation switches. When the temperature dropped below 80 °C, turned off the heating power and fan, we then turned off the power and the cylinder switches, and removed the pressure control valve.

### 4.6. Chromatographic Conditions

We weighed precisely 10mg of dried Cap-NO·H_2_O, added 1ml of methanol to dissolve it, diluted it by 10,000 times (10 μg/mL), and then filtered it with a 25 μm filter into an automatic sample bottle. The chromatographic separation was performed at an HPLC instrument (2695-2489, Waters, USA) equipped with an ODS column (250 mm × 4.6 mm × 5 μm) at the wavelength of 215 nm. Column temperature was set at 40 °C, and the injection volume was 20 μL. Isocratic elution was run for 20 min, and the mobile phase consisted of 0.1% phosphoric acid aqueous solution and methanol (50/50, *v*/*v*) at a flow rate of 1 mL/min.

### 4.7. The Optimization of Cap-NO·H_2_O Storage Conditions by Response Surface Method

#### 4.7.1. Screening Key Factors by Plackett–Burman Design (PBD)

The PB experiment in this paper adopted a 3-level experimental design, therefore n = 3. It adopted an experimental design table with n = 12 times, and the number of dummy variables was 6. It measured the percentage of remaining amount of Cap-NO·H_2_O after 180 days of storage under the control of other variables, such as light, deoxidizer, nitrogen purity, temperature and desiccant. The factor design was shown in Table 5. After screening several significant factors, the key factors were selected in the follow-up experiments.

Exposure experiment: one set of packaging bags was made of brown packaging bags and wrapped in light-proof paper, and the other set was made of transparent packaging bags; deoxidizer experiment: 1.5 g deoxidizer was put into one group before the package was sealed, and no deoxidizer was placed in the other group; nitrogen filling experiment: one group was filled with nitrogen in the package, the other group was filled with air; temperature experiment: one group was placed in a refrigerator at −20 °C, and the other group was exposed to room temperature; desiccant experiment: 1.2 g desiccant was put into one group, and no desiccant was placed in the other group. Each experimental group was repeated 3 times.

#### 4.7.2. The Experiment Design of the Path of Steepest Ascent

According to the PB experiment, after removing the non-critical factors, in order to quickly find the best state point interval of the remaining key factors, the path of steepest ascent was adopted. The first-order linear fitting formula of the steepest climbing experiment was shown as follows:(4)$$\hat{y}=b_{0}+\sum_{i=1}^{k}b_{i}x_{i}$$
where:

$\hat{y}$—Optimal response value;

$b_{i}$—Variation coefficient of influence factor;

$x_{i}$—Impact factors.

The step length formula was shown as follows:(5)$$\Delta x_{i}=\frac{b_{i}}{b_{j}/\Delta x_{j}}$$
where:

$\Delta x_{j}$—The variable step with the largest absolute value of the regression coefficient;

$b_{j}$—Absolute maximum regression coefficient;

$b_{i}$—Calculation coefficient.

#### 4.7.3. The Experiment Design of Response Surface

The response surface method (RSM) is a classical statistical modeling technique dedicated to evaluating the interaction between a single experimental group factor and one or more experimental group factors. Therefore, in order to obtain a reliable model, each factor needs to first be tested. In this paper a central composite experimental design [19] was adopted to optimize the storage conditions of Cap-NO·H_2_O with nitrogen purity, storage temperature, deoxidizer quality, desiccant quality, and light as the influencing factors. Preliminary studies have shown that nitrogen purity, storage temperature and desiccant were important factors affecting the stable storage of Cap-NO·H_2_O. Therefore, these variables were set as *X*_1_, *X*_2_, and *X*_3_, respectively. The variables followed the formula below:(6)$$x_{i}=\frac{\left(X_{i}-{\bar{X}}_{i}\right)}{\Delta X_{i}}\;\;\;\;i=1,2,3,\ldots k,$$
where:

$x_{i}$—Independent variable;

$X_{i}$—The real value of the independent variable;

${\bar{X}}_{i}$—The central value of the independent variable;

$\Delta X_{i}$—Step length.

The independent variable for Box–Behnken factor level design in this paper was shown in Table 6.

The maximum storage capacity of Cap-NO·H_2_O was adopted as the response value ${\hat{Y}}_{i}\left(U\right)$. Regression analysis was carried out on the acquired data by using Design Expert 10. The predicted response was calculated with the following second-order polynomial:(7)$${\hat{Y}}_{i}=\beta_{0}+\sum \beta_{i}D+\sum \beta_{ij}x_{i}x_{j}+\sum \beta_{ii}x_{i}{}^{2}$$
where:

${\hat{Y}}_{i}$—Predicted response value;

$x_{i},x_{j}$—Independent variables;

$\beta_{i}$—Linear coefficient;

$\beta_{ij}$—Interaction term coefficient;

$\beta_{ii}$—Square coefficient.

### 4.8. Statistical Methods

Statistical calculations in this paper used the statistical software Design Expert 10 and IBM SPSS Statistics 23.

## Figures and Tables

**Figure 1 molecules-26-07533-f001:**
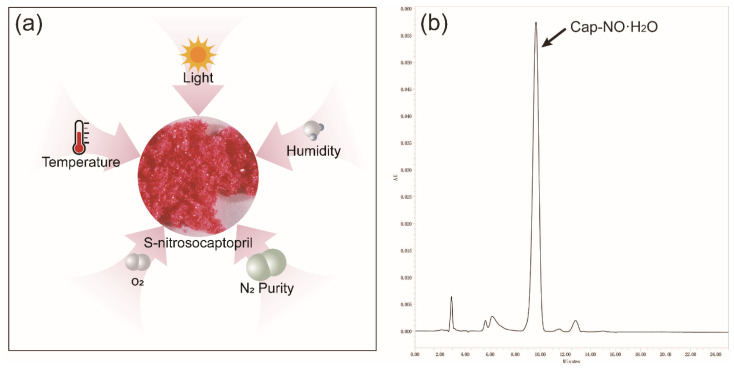
Response test: (**a**) investigated factors and (**b**) HPLC analysis of Cap-NO·H_2_O. The retention time for Cap-NO·H_2_O was 9.65 min.

**Figure 2 molecules-26-07533-f002:**
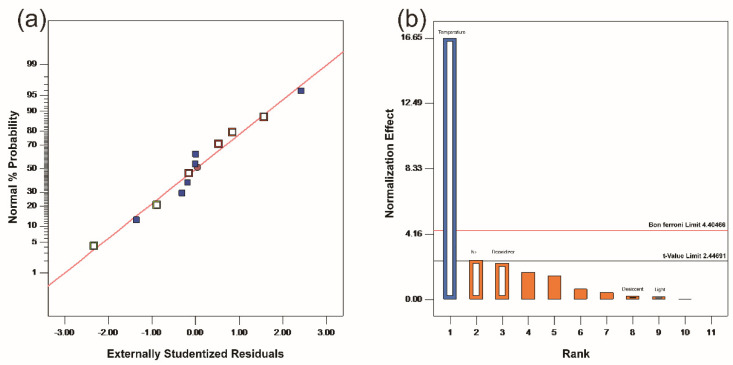
PB experiment of Cap-NO·H_2_O: (**a**) normal plot of residuals; (**b**) Pareto chart (Alpha = 0.05).

**Figure 3 molecules-26-07533-f003:**
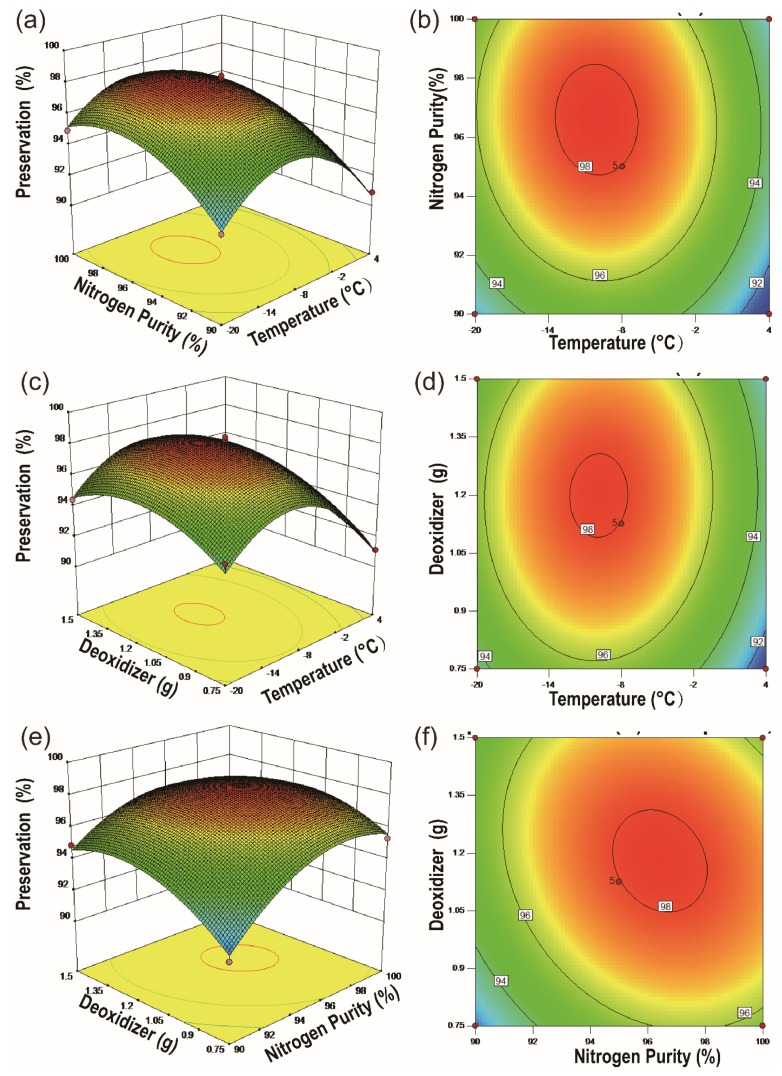
Response surface of Cap-NO·H_2_O preservation (temperature, nitrogen purity): (**a**) surface plot; (**b**) contour plot. Response surface of Cap-NO·H_2_O preservation (temperature, deoxidizer): (**c**) surface plot; (**d**) contour plot. Response surface of Cap-NO·H_2_O preservation (nitrogen purity, deoxidizer): (**e**) surface plot, (**f**) contour plot.

**Figure 4 molecules-26-07533-f004:**
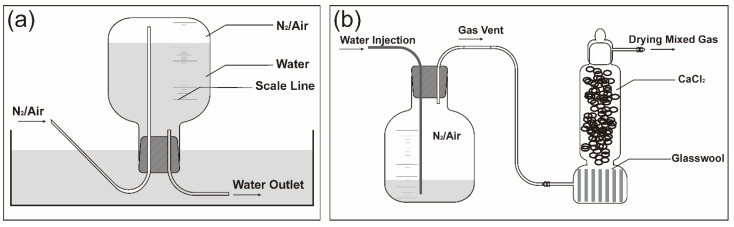
Nitrogen ratio regulating device: (**a**) gas collecting; (**b**) improved air delivery connection device.

**Table 1 molecules-26-07533-t001:** Results of Plackett–Burman experiment.

N						Response
	*X*_1_ Light	*X*_2_ Deoxidizer	*X*_3_ Nitrogen Purity	*X*_4_ Temperature	*X*_5_ Desiccant	Preservation Rate %
1	−1	−1	−1	−1	−1	0.81
2	1	−1	−1	1	1	66.88
3	−1	−1	1	1	1	88.45
4	1	1	1	1	−1	93.38
5	−1	1	1	−1	−1	5.98
6	1	1	−1	1	−1	90.59
7	−1	1	1	1	1	97.82
8	1	−1	1	−1	−1	3.18
9	−1	1	−1	−1	1	1.20
10	−1	−1	−1	1	−1	59.87
11	1	−1	1	−1	1	4.27
12	1	1	−1	−1	1	1.33

**Table 2 molecules-26-07533-t002:** Result of path of steepest ascent.

Step Length	Temperature °C	Nitrogen Purity %	Deoxidizer g	Preservation Rate %
0	28	80	0.00	61.24 ± 0.36
0 + 1Δ	16	85	0.38	82.88 ± 0.46
0 + 2Δ	4	90	0.75	90.45 ± 0.20
0 + 3Δ	−8	95	1.13	97.93 ± 0.66
0 + 4Δ	−20	100	1.50	96.23 ± 0.30

**Table 3 molecules-26-07533-t003:** Results of Box–Behnken response surface design.

Number	Temperature *X*_1_	Nitrogen Purity *X*_2_	Deoxidizer *X*_3_	Preservation Rate %*Y*
1	−1 (−20 °C)	−1 (90%)	0 (1.13 g)	92.19
2	1 (4 °C)	−1	0	90.87
3	−1	1 (100%)	0	94.94
4	1	1	0	92.64
5	−1	0 (95%)	−1 (0.75 g)	94.04
6	1	0	−1	91.12
7	−1	0	1 (1.50 g)	94.43
8	1	0	1	92.08
9	0 (−8 °C)	−1	−1	91.55
10	0	1	−1	95.32
11	0	−1	1	94.88
12	0	1	1	95.99
13	0	0	0	98.27
14	0	0	0	98.43
15	0	0	0	97.90
16	0	0	0	97.55
17	0	0	0	97.77

**Table 4 molecules-26-07533-t004:** Predicted value and measured value.

Group	Temperature °C	Nitrogen Purity %	Deoxidizer g	Preservation Rate %	Expectation
Predicted Value	−10.042	96.480	1.176	98.296	0.983
Measured Value	−10.0	97.0	1.20	97.91 ± 0.59	

**Table 5 molecules-26-07533-t005:** Plackett–Burman factor level table.

Number	Factors	Level	
		−1	+1
*X* _1_	Light	Exposure	Non-exposure
*X* _2_	Deoxidizer	None	Yes
*X* _3_	Nitrogen	Air	N_2_
*X* _4_	Temperature	−20 °C	20 °C
*X* _5_	Desiccant	None	Yes

**Table 6 molecules-26-07533-t006:** Box–Behnken factor level design.

Factor	Level		
	−1	0	1
Temperature	−20	−8	4
Nitrogen purity	90	95	100
Deoxidizer	0.75	1.13	1.50

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
