# Peer review of "Optimization of S-Nitrosocaptopril Monohydrate Storage Conditions Based on Response Surface Method"

_molecules, 2021, doi:10.3390/molecules26247533_

Round 1

Reviewer 1 Report

Dear Authors,

Thank you for the opportunity of reviewing your manuscript entitled: ”Optimization of monohydrate S-nitrosocaptopril storage conditions based on response surface method”.

After reading the manuscript I have a couple of comments:

Major concerns:

  1. In the presented studies, the authors use the factor (the variable ) named “nitrogen”. But, in the opinion of the reviewer the problem of the degradation is caused by the oxygen. Please consider to change the factor or give explanations in text. Indeed the concentration of the nitrogen in the gas mixture is a different method to determining the oxygen concentration. It should be clarified for all readers.
  2. Line 205 – Is used packaging bags protect from the gas exchange? Is the composition of the gas inside of the bags doesn’t changed during 180 days tests?
  3. Line 246-247 – in text, we have two times the same factor “storage temperature and storage temperature” Please correct.
  4. Line 257, Tabel 2 and 6 – Accordingly to the response surface methodology the step length between the factor levels should be equal. In the case of factor “desiccant” we have two different step lengths 0.38 for levels -1 and 0, and 0.37 for levels 0 and +1. Please check and confirm the real values of the independent variable.
  5. What type of packaging material was used in the optimization experiments?
  6. Line 327 - 330 " Under the interaction … under the interaction " is not clear. Please rewrite more clearly.
  7. Line 336 - 343 " However, the equation had…. … , and the Y was 96,4796 " is not clear. Please rewrite more clearly
  8. Discussion – line 372 to 388 – The response surface experiments were conducted for desiccant. The discussion is about the deoxidizer. If I understand properly, no desiccant was used in the optimization experiments. Please correct the discussion or the results.

Minor concerns:

  1. Accordingly to “instructions for authors” the structure of the manuscript is inappropriate. Chapters should be placed in order: Introduction, Results, Discussion, Materials and Methods.
  2. The quality of figure 1. Please improve the labels in it.
  3. Text quality:
  • before the square brackets of the reference numbers should be placed “one space”,
  1. Reference:
  • To all citations - follow the rules of the “instructions for authors”. Please use Abbreviated Journal Name and include the digital object identifier (DOI) for all references where available.
  • Citation no 10– Please, correct title
  • Citation no 11– Please, correct article title
  • Citation no 16– Please, check the citation. The reviewer couldn’t find any internet information about this article.
  • Citation no 17– Please, correct handbook title
  • Citation no 20– Please, correct year of publication

The paper is ready for publication in Molecules after revision.

Reviewer 2 Report

The manuscript by Huang et al. described the optimal storage conditions of S-nitrosocaptopril monohydrate for its long-term storage using Response surface methodology (RSM). S-nitrosocaptopril monohydrate stability is crucial for studying its medicinal properties as this compound has implication for the treatment of various diseases.

The experiments described in this paper are technically sound and the results are discussed appropriately in the context of the existing literature.

I think a few issues need clarification to make for a stronger, and perhaps more generally appealing paper, as follows:

  1. In figure 3b, what other peaks in the HPLC chromatograph corresponds to? Is this the chromatograph of purified Cap-NO-H2O?
  2. At page 13, line 370, it has been mentioned that desiccant had no significant influence on storage.. After preparation of the compound how it was stored prior to the analysis of storage conditions and for how long?

Reviewer 3 Report

Molecules-MDPI

Manuscript number: molecules-1480855

Title: Optimization of Monohydrate S-nitrosocaptopril Storage Conditions Based on Response Surface Method

The manuscript “Optimization of Monohydrate S-nitrosocaptopril Storage Conditions Based on Response Surface Method“ is a paper that reports the statistical optimization of monohydrate S- nitrosocaptopril conditions for storage. The authors performed Placket Burman design and after that Box-Benken design.

This paper is not very well written. Although the whole idea is interesting, a lot of information is missing but on the other hand, a lot of superfluous information about statistics (Plackett-Burman design) was presented. Also, additional experiments must be done in this paper. The main criticism is reflected in shortcomings of explanation regarding response surface methodology/statistical optimization. In the title of the manuscript, the words „Response Surface Method” are present but the essence of the manuscript is not discussed and presented well.

Major comments:

  1. Page 4, line 130…. 78%x+y ………..Is this equation in a Table or something else? Please correct this
  2. Page 5, Screening factors for Plackett-Burman design (PBD), lines 173-204…PBD is already known in the literature, there a lot of papers dealing with PBD, thus some parts of the text (lines 173-204) are superfluous- for example, calculation of the experimental error, standard deviation, t-test and p-value….Please correct this.
  3. Page 6, The experiment design of the path of steepest ascent, lines 215-228 are superfluous, also as Figure 2. In this paragraph (lines 215-228) the authors are explaining how the PB design is working…..what is the lack of fit….this is not for the Materials and method section…The PBD is a statistical tool that you are using for achieving your goal. Please correct this.
  4. This manuscript has 8 tables. Please remove Table 4 and Table 7 from the Manuscript body and put them in the Supplementary Material.
  5. Page 7, The experiment design of response surface, line 255- Why you used Box-Benken design? Why not some other design, Central Composite Design (CCD) or Optimal design? Please explain this.
  6. Page 13, Discussion section- The Discussion section is very poor.
  7. The authors gave results on page 10 and 11 that are mainly statistic data (Design Expert presents statistic data for each experiment). What about Figure 6? Could you please explain surface plots a), c) and e)? What about interactions between components nitrogen –temperature (a), desiccant- temperature (b) and desiccant –temperature (c) (Figure 6)? This is a crucial thing in optimization. Please add these explanations in the Discussion section
  8. Line 350 “Contour plot and 3D surface of temperature and nitrogen content”-please explain why this sentence is under Figure 6?
  9. Validation of the model is missing. When you validate a model, please include PI intervals.

Minor comments:

Page 2, Introduction section, lines 80-82, sentence “Then, response surface methodology (RSM) should be conducted to maximize the extraction yield as a statistical multi response optimization method[14].”- what extraction yield? Please correct this

This paper has a lot of mistakes, it is not well organized and written. There is a lot of superfluous information and unfortunately less information that are the essence. Thus, this paper is not for publication in Molecules.

Round 2

Reviewer 1 Report

Dear Authors,

You modified the manuscript accordingly to the reviewer’s suggestion, and in my opinion, your manuscript can be accepted for publication in Molecules.

Best regards,

Reviewer 3 Report

The authors answered all questions previously posted by reviewers. Now, the manuscript is improved in several aspects including the discussion section. 

In this form, this paper is suitable for publication.